# Competitive Coordination-Oriented Monodispersed Cobalt Sites on a N-Rich Porous Carbon Microsphere Catalyst for High-Performance Zn−Air Batteries

**DOI:** 10.3390/nano13081330

**Published:** 2023-04-10

**Authors:** Mengxia Shen, Hao Yang, Qingqing Liu, Qianyu Wang, Jun Liu, Jiale Qi, Xinyu Xu, Jiahua Zhu, Lilong Zhang, Yonghao Ni

**Affiliations:** 1College of Bioresources Chemical and Materials Engineering, Key Laboratory of Auxiliary Chemistry and Technology for Chemical Industry, Ministry of Education, Shaanxi Collaborative Innovation Center of Industrial Auxiliary Chemistry and Technology, Shaanxi University of Science and Technology, Xi’an 710021, China; 2State Key Laboratory of Material-Oriented Chemical Engineering, College of Chemical Engineering, Nanjing Tech University, Nanjing 211816, China; 3Department of Chemical Engineering, University of New Brunswick, Fredericton, NB E3B 5A3, Canada; 4Department of Chemical and Biomedical Engineering, University of Maine, Orono, ME 04469, USA

**Keywords:** oxygen reduction reaction, single-atom catalysts, nitrogen-rich, supramolecular coordination, Zn−air battery

## Abstract

Metal/nitrogen-doped carbon single-atom catalysts (M−N−C SACs) show excellent catalytic performance with a maximum atom utilization and customizable tunable electronic structure. However, precisely modulating the M−N_x_ coordination in M−N−C SACs remains a grand challenge. Here, we used a N-rich nucleobase coordination self-assembly strategy to precisely regulate the dispersion of metal atoms by controlling the metal ratio. Meanwhile, the elimination of Zn during pyrolysis produced porous carbon microspheres with a specific surface area of up to 1151 m^2^ g^−1^, allowing maximum exposure of Co−N_4_ sites and facilitating charge transport in the oxygen reduction reaction (ORR) process. Thereby, the monodispersed cobalt sites (Co−N_4_) in N-rich (18.49 at%) porous carbon microspheres (CoSA/N−PCMS) displayed excellent ORR activity under alkaline conditions. Simultaneously, the Zn−air battery (ZAB) assembled with CoSA/N−PCMS outperformed Pt/C+RuO_2_-based ZABs in terms of power density and capacity, proving that they have good prospects for practical application.

## 1. Introduction

Developing efficient oxygen reduction reaction (ORR) electrocatalysts is the leading research direction for electrochemical energy devices, including fuel cells and metal-air batteries [1,2,3,4,5]. Currently, platinum-based materials have been widely used as ORR catalysts, but their instability and prohibitive cost inevitably hinder large-scale applications [6,7,8]. Metal/nitrogen-doped carbon (M−N−C) catalysts have attracted increasing interest as a promising substitute for noble-metal-based catalysts, ascribed to their advanced catalytic performance, facile synthesis and low cost [9,10,11,12]. However, M−N−C catalysts are affected by the aggregation of active components during intense pyrolysis, resulting in poor utilization of the M−N_x_ sites [13,14].

M−N−C single-atom catalysts (M−N−C SACs) have been proposed to substantially advance the number of available M−N_x_ sites and realize the maximum atomic utilization [15,16,17,18,19]. Simultaneously, achieving a rational catalyst design at the atomic scale can additionally boost the intrinsic activity of M−N_x_ by modulating its electronic configuration [20]. In most cases, M−N−C SACs are usually prepared by pyrolysis of metal salts, carbon supports and precursors comprising nitrogeN−Containing small molecules [21,22]. To avoid the uncontrolled formation of metal crystals confined in the carbon substrate, post-treatment with highly corrosive acids (e.g., nitric acid and hydrochloric acid) under extremely rigorous conditions is required [23]. However, slight changes in the manipulation strategy may gravely disrupt the electronic structure and coordination of the M−N_x_ sites, which is also undesirable for large-scale production [24].

The well-defined metal-organic frameworks (MOFs) and other supramolecular coordination composites can potentially be used as desirable precursors for SACs by altering the metal nodes and organic ligands [25,26]. For example, by regulating the molar ratio of Zn and Co in ZIF-8, the competitive interaction of the Zn ions with the N atoms in the framework effectively widens the spatial separation between the two Co atoms to prevent their aggregation, while the sublimation of Zn above 907 °C contributes to the porous structure [27,28,29,30]. Nevertheless, the ligand (2-methylimidazole) of ZIF-8 contains a limited nitrogen content, usually leading to insufficient generation of M−N_x_ sites. Therefore, incorporating additional nitrogen-rich sources (melamine, urea, ammonia, etc.) into the synthetic approach has been widely reported, but is considered a relatively cumbersome solution, which probably leads to buried M−N_x_ sites in the SACs [31,32,33].

In this study, adenine−Zn/Co supramolecular coordination microspheres (Ad−Zn/Co SCMS) were successfully prepared by a competitive coordination strategy. Adenine, as a nitrogen-rich (51.8 wt%) ligand, provides a large amount of reaction sites for metal ions. The creation of monodispersed cobalt sites (Co−N_4_) on N-rich porous carbon microspheres (CoSA/N−PCMS) after pyrolysis was ensured by rationally adjusting the Zn/Co−N coordination and Zn/Co ratio in the Ad−Zn/Co SCMS precursor. Meanwhile, the elimination of Zn during pyrolysis produced porous carbon microspheres with a specific surface area up to 1151 m^2^ g^−1^, allowing maximum exposure of Co−N_4_ sites and facilitating charge transport for the ORR process. The ultra-high N doping (18.49 at%) promoted the efficient utilization of Co single atoms, which could significantly improve the activity of ORR [34,35,36,37,38]. The onset potential (E_onset_) and half-wave (E_1/2_) potential of CoSA/N−PCMS under alkaline conditions are 0.99 V and 0.87 V, respectively. Furthermore, CoSA/N−PCMS as a cathode material for Zn-air batteries (ZAB) exxhibits high opeN−Circuit voltage, power density, specific capacity, and charge/discharge cycle stability.

## 2. Experimental Section

### 2.1. Chemicals

Adenine was received from Sigma-Aldrich (Darmstadt, Germany). Pt/C (20 wt%) and RuO_2_ were purchased from Alfa Aesar (Shanghai, China). N, N-dimethylformamide (DMF), Co(NO_3_)_2_·6H_2_O, and Zn(NO_3_)_2_·6H_2_O were obtained from Macklin (Shanghai, China). Nafion solution (5%) was obtained from Macklin.

### 2.2. Preparation of Ad−Zn/Co SCMS Precursor and CoSA/N−PCMS Catalysts

Adenine (1.08 g) was dissolved in 400 mL DMF at 140 °C. Zn(NO_3_)_2_·6H_2_O (3.996 mmol) and Co(NO_3_)_2_·6H_2_O (0.004 mmol) were added to the adenine solution, which was stirred continuously for 4 h. After the reaction, Ad−Zn/Co SCMS was collected by centrifugation and then dried in an oven. CoSA/N−PCMS was obtained by directly calcining Ad−Zn/Co SCMS in a tube furnace at 1000 °C for 2 h under an Ar atmosphere (ramping rate of 5 °C min^−1^).

For comparison, the preparation method of Ad−Zn SCMS and Ad−Co SCMS is similar to that of Ad−Zn/Co SCMS, with the exception that only 0.4 mmol of Co(NO_3_)_2_·6H_2_O or Zn(NO_3_)_2_·6H_2_O needs to be added. The calcined products of Ad−Co SCMS and Ad−Zn SCMS are denoted as Co/N−PCMS and N−PCMS, respectively.

### 2.3. Characterization

Scanning electron microscopy (SEM, SU8100, HITACHI, Tokyo, Japan) and transmission electron microscopy (TEM, FEI Tecnai G2 F20 S-TWIN, FEI, Hillsboro, OR, USA) were applied to observe the morphology of the samples. The images of annular bright-field scanning transmission electron microscopy (ABF-STEM) and aberratioN−Corrected high-angle dark-field scanning transmission electron microscopy (HAADF-STEM) of CoSA/N−PCMS were collected on a JEM-GRAND ARM300F (JEOL, Tokyo, Japan). X−ray diffraction (XRD) spectroscopy of samples was performed on a Bruker D8 Advance A25 diffractometer (Bruker, Karlsruhe, Germany). The surface valence state and elemental composition of the catalyst were obtained by XPS (AXIS Supra, Kratos, Manchester, UK). The Brunauer-Emmett-Teller (BET) specific surface area and pore size distribution were acquired on a gas analyzer ASAP 2460 (Micromeritics, Norcross, GA, USA). A Raman spectrometer (DXRxi, Thermo-Fisher, Waltham, MA, USA) was used to determine the extent of defects of the catalysts. X−ray absorption spectroscopy (XAS) of the catalysts was performed on the 1W1B beamline at BSRF (Beijing Synchrotron Radiation Facility, Beijing, China).

### 2.4. Electrochemical Measurements

ORR electrocatalytic tests were performed in a three-electrode system with platinum foil and Ag/AgCl as the counter and reference electrodes, respectively. All tests were performed on a CHI 760E electrochemical workstation. The catalyst sample (5 mg), Nafion (5%, 10 μL), and ethanol (490 μL) were mixed, then sonicated for more than 1 h to obtain a homogeneous ink. Then, 8 μL or 10 μL of ink was dipped onto the rotating disk electrode (RDE, geometric area ≈ 0.20 cm^2^) or rotating ring disk electrode (RRDE, geometric area ≈ 0.25 cm^2^) surface with a loading of 0.4 mg cm^−2^. Linear scanning voltammetry (LSV) and cyclic voltammetry (CV) were carried out in O_2_-saturated 0.1 M KOH solution at a scanning rate of 10 mV s^−1^. In addition, for the accelerated durability test (ADT), the RDE was subjected to potential cycling. Specifically, the electrode was scanned from −0.4 to 0 V at a rate of 100 mV s^−1^. CV and ORR curves were recorded at each set of 1000 (1 k) cycles.

The potential measured by Ag/AgCl can be converted to the potential of the reversible hydrogen electrode (RHE) by the following equation:E_RHE_ = E_Ag/AgCl_ + 0.059 pH + 0.197(1)

The kinetic current density (*J_k_*) and electron transfer number (*n*) were calculated from the Koutecky-Levich (K-L) equation:(2)$$\frac{1}{J}=\frac{1}{J_{k}}+\frac{1}{J_{d}}=\frac{1}{J_{k}}+\frac{1}{B\omega^{1/2}}$$
(3)$$B=0.2nFC_{0}D^{2/3}\nu^{-1/6}$$
in which *ω* is the electrode rotation rate, *J_k_* is the kinetic current density, *J_d_* is the diffusion-limiting current density, *J* is the measured current density, *B* is the inverse of the slope of K-L equation, *F* is the Faraday constant, *C*_0_ is the oxygen concentration in 0.1 M KOH solution, *D* is the diffusion coefficient of O_2_, and *ν* is the kinematic viscosity of the electrolyte.

The kinetic current was calculated using the following equation:(4)$$J_{k}=\frac{J\times J_{d}}{J_{d}-J}$$

The hydrogen peroxide yield (H_2_O_2_%) and the number of electron transfers (*n*) were calculated using the following equations:(5)$$H_{2}O_{2}\left(\%\right)=200\times \frac{I_{r}/N}{{I_{d}+I}_{r}/N}$$
(6)$$n=4\times \frac{I_{d}}{{I_{d}+I}_{r}/N}$$
where *I_r_* is the ring current, *I_d_* is the disk current, and *N* = 0.37 is the collection efficiency.

### 2.5. Fabrication and Measurements of Zn−Air Battery

In the Zn−air battery test, a polished Zn sheet (8 cm × 3.2 cm) was used as the anode, a carbon cloth coated with CoSA/N−PCMS or Pt/C+RuO_2_ (loading: 2 mg cm^−2^) was used as the cathode (air electrode), and the electrolyte consisted of a mixture of 6 M KOH/0.2 M zinc acetate. The battery was tested using a CHI 760E electrochemical workstation.

## 3. Results and Discussion

The synthetic route for CoSA/N−PCMS is illustrated in Figure 1a. As one of the nucleobases that is present in nucleic acids, adenine (N content up to 51.8 wt%) contains a lone pair of electrons at the nitrogen atom and strongly coordinates with 3D metals (Zn(II) and Co(II)), which triggers the self-assembly of adenine−Zn/Co supramolecular coordination microspheres (Ad−Zn/Co SCMS) [39,40]. By rationally adjusting the Zn/Co ratio, excess incorporated Zn ions occupy most of the coordination nodes in the Ad−Zn/Co building blocks, which guarantees the isolation of atomically dispersed Co sites in the pyrolyzed products. As reflected in the SEM image (Figure 1b), Ad−Zn/Co SCMS shows monodisperse micro-spherical features with an average diameter of about 1 μm. With the aid of an elemental mapping analysis (Figure 1c), both Co and Zn were determined to be uniformly distributed throughout the microspheres. As controls, Ad−Zn SCMS and Ad−Co SCMS were synthesized by employing Zn(II) or Co(II), respectively, as the sole metal source (Appendix A). The red shift in the UV-vis absorption peaks of Ad−Zn/Co, Ad−Zn, and Ad−Co SCMS with respect to adenine indicates the coordination between Ad and the metal ions (Appendix A). From the FTIR spectra analyses (Appendix A), the adenine molecule has the characteristic stretching and bending vibrations of an imidazole N at 1120 cm^−1^ and the stretching vibrations of an amine (-NH_2_) in the band range of 3300−3100 cm^−1^, which are noticeably attenuated in the assembled Ad−Zn/Co SCMS, Ad−Zn SCMS, and Ad−Co SCMS.

Furthermore, XAS was used to characterize the chemical states and local structure of Co and Zn in Ad−Zn/Co SCMS. From the X−ray absorption near-edge structure (XANES) spectra (Appendix A), the near-edge absorption energy of Co or Zn moves towards the high-energy region, indicating that Co or Zn exists in an oxidized state. More structural information about Co and Zn can be obtained from the extended X−ray absorption fine structure (EXAFS). The Co K-edge Fourier transform (FT) EXAFS spectrum (Figure 1d) of Ad−Zn/Co SCMS exhibits similar Co−N coordination with a peak located at ~1.41Å. Meanwhile, the Zn K-edge FT-EXAFS spectrum (Figure 1e) of Ad−Zn/Co SCMS displays a main peak at ~1.56 Å, corresponding to the Zn−N scattering path [40,41,42,43]. Therefore, both Co and Zn coordinate with N on adenine, confirming the feasibility of achieving Co atomic dispersion by precisely regulating the metal ratio.

CoSA/N−PCMS exhibits regular microspheres with uniform distribution of C, N, and Co (Figure 2a and Appendix A). Benefitting from the use of N-rich adenine as the ligand, the nitrogen atom loading content in CoSA/N−PCMS was as high as 18.49 at% (Appendix A). To further investigate the microstructure of the catalyst, high-resolution TEM (HRTEM) was used, and rough surface and edge defects on the CoSA/N−PCMS can be clearly seen in the image (Figure 2b). Meanwhile, the ring-like selected area electron diffraction (SAED, illustration of Figure 2b) pattern indicates that the CoSA/N− PCMS catalyst has no metallic Co. Furthermore, only amorphous carbon and lattice deformation were observed in the ABF-STEM image (Figure 2c), further confirming the absence of Co aggregation. In comparison, large, aggregated Co nanoparticles were found in the Co/N−PCMS without Zn addition (Appendix A), confirming that the precise modulation of metals in Ad−Zn/Co SCMS precursors is a decisive factor in achieving the dispersion of Co atoms. The crystal structures of the catalysts were further investigated by XRD. There are only broad typical diffraction peaks in the (101) and (002) planes of amorphous carbon in N−PCMS and CoSA/N−PCMS (Figure 2d), while Co/N−PCMS has distinct Co metal peaks (Appendix A) [41]. All of the above results confirm that the high dispersion of Co species was confined in CoSA/N−PCMS. Raman spectra (Figure 2e) were measured to characterize the carbon structure in the catalysts. Compared with N−PCMS and Co/N−PCMS, CoSA/N−PCMS has the largest I_D_/I_G_ ratio, related to the promotion of defects by Co doping and Zn volatilization. Then, the porosity character of the CoSA/N−PCMS was investigated using N_2_ physisorption measurements (Figure 2f) [42]. The amount of absorbed N_2_ increases dramatically at relatively low pressures of 0−0.005, which can be ascribed to the generation of many micropores by volatilization of Zn during the pyrolysis process. CoSA/N−PCMS possesses a high specific surface area of 1151 m^2^ g^−1^, surpassing that of Co/N−PCMS (98 m^2^ g^−1^). According to the pore size distribution curves (Appendix A), CoSA/N−PCMS has an abundance of micropores [43]. The micropore-dominated structure and large specific surface area could not only host a high density of active sites but also provide sufficient channels to sufficiently boost the mass transport in electrocatalysis [44,45].

To further reveal the spatial location of Co atoms in CoSA/N−PCMS, HAADF-STEM was conducted. As shown in Figure 3a, the atomic bright spots are circled in yellow, confirming that the atoms of Co are dispersed in CoSA/N−PCMS [46]. Then, XPS was used to elucidate the chemical composition of CoSA/N−PCMS, in which Co, C, N, and O elements all exist (Appendix A). The N 1 s spectrum of CoSA/N−PCMS (Figure 3b) was assigned to pyridine nitrogen (398.4 eV), pyrrole nitrogen (399.5 eV), graphitic nitrogen (401.0 eV), and oxidized nitrogen (402.4 eV) with the Casaxps software analysis. The presence of pyrrole and pyridine nitrogens provides sufficient anchoring points for the single metal sites in the M−N−C catalyst, which also indicates the successful coordination of N−Co in the catalyst for facilitating the ORR performance [47,48,49,50,51]. The Co K-edge XANES spectra of different cobalt-containing samples are shown in Figure 3c. The Co near-edge absorption position of CoSA/N−PCMS is located between the Co foil and CoPc, indicating that the Co atoms are positively charged [52]. In contrast, the pre-edge peak of Co/N−PCMS is similar to that of Co foil, confirming the possible generation of metallic Co. As reflected in Figure 3d, a dominant peak at 2.18 Å attributed to Co−Co binding in Co/N−PCMS further confirms the existence of metallic Co. The FT-EXAFS curve (Figure 3d) for CoSA/N−PCMS only shows a major peak at 1.41 Å, originating from the Co−N configuration. Meanwhile, no obvious Co−Co peaks were found, indicating that Co is dispersed atomically in CoSA/N−PCMS [53]. These observations agree with the conclusions of XRD, SAED, and HAADF-STEM. The detailed quantitative parameters of CoSA/N−PCMS were obtained by EXAFS fitting (Appendix A and Appendix A), and the Co−N coordination number is 4, with a bond length of 1.89 Å. The above results indicate that Co atoms in CoSA/N−PCMS exist in the form of Co−N_4_ [53,54,55]. Nucleobase-engaged competitive coordination self-assembly is also applicable to other transition metals, such as Fe, indicating the universality of this approach (Appendix A). As a result, CoSA/N−PCMS with regular microspheres, atomically dispersed Co−N_4_ sites, a high content of N, and a porous structure was prepared by precise regulation. These features will assist in enhancing the performance of the ORR [56,57].

The cyclic voltammetry (CV) curves under alkaline conditions are shown in Figure 4a. All four samples showed significant ORR peaks, with CoSA/N−PCMS having the highest peak potential (0.90 V), surpassing Pt/C (0.84 V), N−PCMS (0.84 V), and Co/N−PCMS (0.80 V). The linear sweep voltammetry (LSV) curve of CoSA/N−PCMS (Figure 4b) exhibited the highest catalytic activity compared to Pt/C, N−PCMS, and Co/N−PCMS, owing to the remarkably high onset potential (E_onset_ = 0.99 V) and half-wave potential (E_1/2_ = 0.87 V). Appendix A summarizes the E_onset_ and E_1/2_ of the four samples [58]. Then, the Tafel slopes obtained from LSVs revealed that CoSA/N−PCMS (74 mV dec^−1^) was superior to Pt/C (81 mV dec^−1^), indicating excellent reaction kinetics. To further evaluate the ORR of CoSA/N−PCMS, Koutecky-Levich (K-L) curves (Appendix A) were obtained by performing RDE measurements at varied rotation speeds. The electron transfer number calculated from the K-L plots is about 3.92, indicating a four-electron oxygen reduction process [59]. As shown in Figure 4d, the hydrogen peroxide yield and electron transfer number of CoSA/N−PCMS are similar to those of Pt/C, which is comparable to the values calculated by the K-L equation. The accelerated durability test (ADT) and chronoamperometric response (i−t) were conducted to evaluate the durability of CoSA/N−PCMS [60]. The ADT results (Figure 4e,f) show that after 5000 consecutive cycles, the E_1/2_ of CoSA/N−PCMS has a negative shift of only 9 mV, significantly better than Pt/C (35 mV). After ADT testing, CoSA/N−PCMS was analyzed by XRD to investigate whether metal aggregation occurred. As shown in Appendix A, there were only two broad peaks attributed to carbon before and after the CoSA/N−PCMS test, indicating that no metal aggregation occurred. The chronoamperometric response (i−t) curves for CoSA/N−PCMS and Pt/C at 0.6 V are shown in Appendix A. The current density for CoSA/N−PCMS remains 91.7% for about 15 h, while the Pt/C catalyst only has 80.5% retention. In conclusion, the ORR performance of CoSA/N−PCMS in 0.1 M KOH exceeds that of Pt/C. CoSA/N−PCMS has a large amount of atomically dispersed Co−N_4_ active sites compared to N−PCMS, and possesses a considerably larger specific surface area and abundant micropores in contrast to Co/N−PCMS, thus exhibiting excellent ORR performance. Moreover, CoSA/N−PCMS also has superior catalytic activity compared to those of recently reported Co SACs with less N doping (Appendix A).

To verify the viability of CoSA/N−PCMS in practical applications, a Zn−air battery (ZAB) was assembled (Figure 5a). CoSA/N−PCMS was used as the air cathode, polished zinc flakes were used as the anode, and 6 M KOH and 0.2 M zinc acetate were used as the electrolyte [61]. As displayed in Figure 5b, the CoSA/N−PCMS-assembled ZAB has a higher and more stable OCV of 1.47 V than the ZAB based on Pt/C + RuO_2_ (1.40 V). The ZAB with a CoSA/N−PCMS air cathode exhibits a maximum power density of 168.7 mW cm^−2^, which is better than the ZAB assembled with Pt/C+RuO_2_ (146.3 mW cm^−2^) (Figure 5c). Moreover, its discharge capacity of 796 mA h g^−1^ outperforms that of Pt/C+RuO_2_ (698 mA h g^−1^) at a current density of 10 mA cm^−2^ (Figure 5d). As shown in Figure 5e, static discharge tests were performed on ZABs assembled with CoSA/N−PCMS with the current density of 0−40 mA cm^−2^. The rate performance of ZAB under varied current densities all show very stable voltages, proving the excellent discharge rate performance of the CoSA/N−PCMS-based ZAB. In addition, constant-current charging and discharging were performed to further investigate the stability of the ZAB (Figure 5f). The initial discharge potential of the CoSA/N−PCMS ZAB was 1.17 V and the charging potential was 2.08 V, with a charge−discharge voltage gap of 0.91 V. After 90 h of cycling, the ZAB based on CoSA/N−PCMS maintained a negligible voltage gap change. However, the stability performance of the Pt/C+RuO_2_-based ZAB decreased sharply after 50 h. The above results demonstrate the viability of the practical application of CoSA/N−PCMS in ZABs.

## 4. Conclusions

In summary, highly active Co monoatomic catalysts with a high nitrogen content were successfully prepared by a N-rich nucleobase-engaged competitive coordination self-assembly strategy. The N originating from Ad−Zn/Co SCMS promotes the coordination of Co during pyrolysis. The large specific surface area, abundant micropores, and the wealth of atomically dispersed Co−N_4_ sites result in CoSA/N−PCMS exhibiting excellent ORR properties (E_onset_ = 0.99 V, E_1/2_ = 0.87 V). In particular, a ZAB based on CoSA/N−PCMS exhibits a maximum power density of 168.7 mW cm^−2^, an impressive specific capacity of 796 mA h g^−1^, and excellent cycling durability over 90 h. This work aims to regulate the coordination environment of metal atoms by using a nitrogen-rich molecule to improve the catalytic activity of the M−N−C SACs.

## Figures and Tables

**Figure 1 nanomaterials-13-01330-f001:**
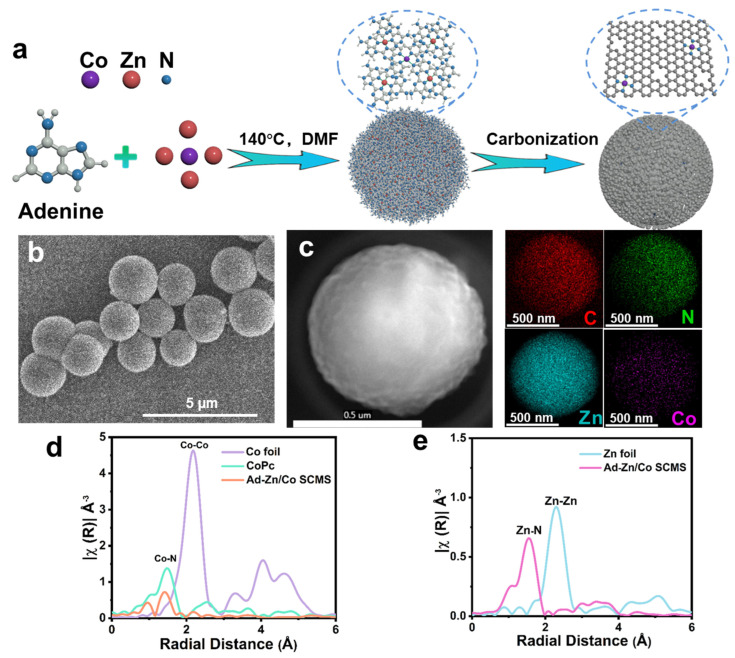
(**a**) Schematic diagram for synthesizing Ad−Zn/Co SCMS and CoSA/N−PCMS. (**b**) SEM image of Ad−Zn/Co SCMS. (**c**) Elemental mapping images of Ad−Zn/Co SCMS. (**d**) Co K-edge and (**e**) Zn K-edge FT-EXAFS spectra of Ad−Zn/Co SCMS and reference samples.

**Figure 2 nanomaterials-13-01330-f002:**
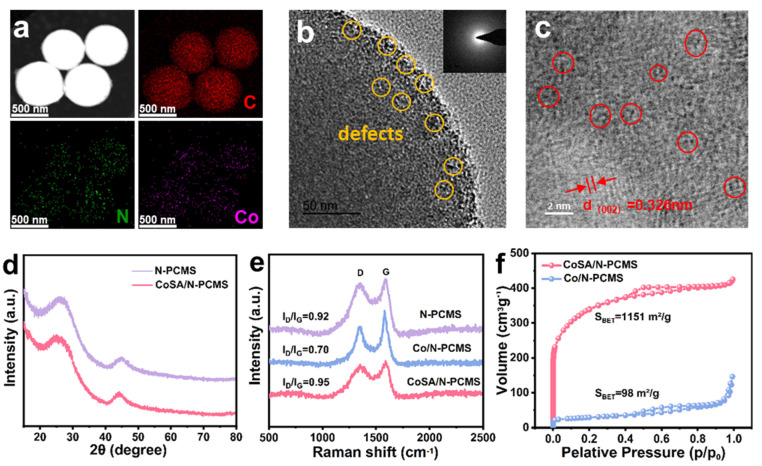
(**a**) Elemental mapping images of CoSA/N−PCMS. (**b**) HRTEM image of CoSA/N−PCMS (inset: corresponding SAED pattern). (**c**) ABF-STEM image of CoSA/N−PCMS. (**d**) XRD patterns and (**e**) Raman spectra of CoSA/N−PCMS and reference samples. (**f**) N_2_ adsorption-desorption isotherms of CoSA/N−PCMS and Co/N−PCMS.

**Figure 3 nanomaterials-13-01330-f003:**
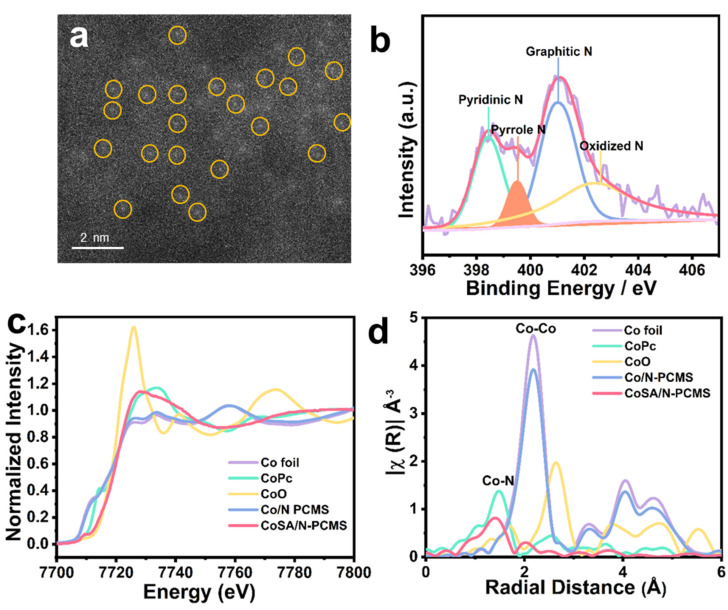
(**a**) HAADF-STEM image of CoSA/N−PCMS with atomically dispersed Co atoms partially circled. (**b**) N 1 s XPS high-resolution spectrum of CoSA/N−PCMS. (**c**) XANES spectra and (**d**) FT-EXAFS spectra of CoSA/N−PCMS and reference samples at the Co K-edge.

**Figure 4 nanomaterials-13-01330-f004:**
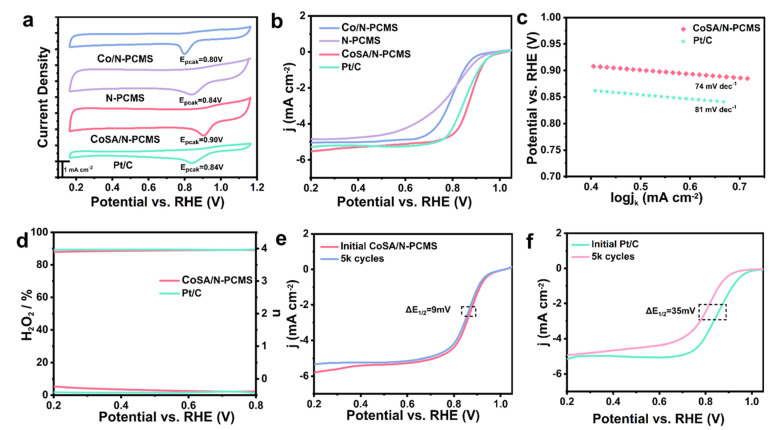
(**a**) CV curves and (**b**) LSV curves of CoSA/N−PCMS, N−PCMS, Co/N−PCMS, and Pt/C. (**c**) Tafel plots and (**d**) electron transfer number (n) and H_2_O_2_ yield of CoSA/N−PCMS and Pt/C. ADT test of (**e**) CoSA/N−PCMS and (**f**) Pt/C for 5000 cycles.

**Figure 5 nanomaterials-13-01330-f005:**
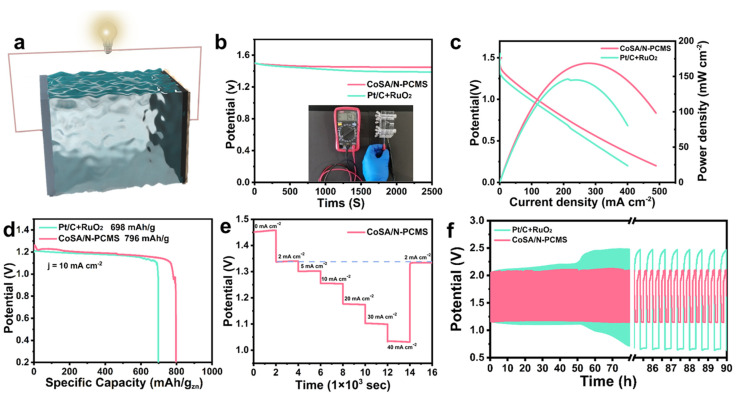
(**a**) Schematic diagram of the aqueous ZAB using CoSA/N−PCMS as an air cathode. (**b**) OpeN−Circuit voltage (OCV) measurements of assembled ZABs with CoSA/N−PCMS and Pt/C+RuO_2_. (**c**) The polarization and power density results for ZABs with CoSA/N−PCMS and Pt/C+RuO_2_. (**d**) The discharge curves of ZABs based on CoSA/N−PCMS and Pt/C+RuO_2_ at 10 mA/cm^2^. (**e**) Discharge curves of the ZAB assembled with CoSA/N−PCMS at varied current densities. (**f**) Discharge/charge cycling performance of ZABs based on CoSA/N−PCMS and Pt/C+RuO_2_ at 10 mA/cm^2^.

## Data Availability

Data will be made available on request.

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
