# Peer review of "Competitive Coordination-Oriented Monodispersed Cobalt Sites on a N-Rich Porous Carbon Microsphere Catalyst for High-Performance Zn−Air Batteries"

_nanomaterials, 2023, doi:10.3390/nano13081330_

Round 1
Reviewer 1 Report
The results obtained are important in the development of efficient catalysts for high-performance Zn-air batteries. The manuscript presents useful results and merits publication.
Reviewer 2 Report
Manuscript ID: nanomaterials-2312932
“Competitive coordination-oriented monodispersed cobalt sites on N-rich porous carbon microspheres catalyst for high-performance Zn-air batteries”, by Mengxia Shen, Hao Yang, Qingqing Liu, Qianyu Wang, Jun Liu, Jiale Qi, Xinyu Xu, Jiahua Zhu, Lilong Zhang, Yonghao Ni
The work is of good quality and publishable after addressing the major comments mentioned below.
Comments to the authors:
1. Page 1, line 24 (Abstract). “Thereby, the monodispersed cobalt sites (Co-N4) on N-rich (18.49 at%) porous carbon microspheres” Is this extremely high N-doping level (18.49 at%) used in the present work really needed? There is a threat that the electrical conductivity of carbon material decreases for such high doping amount of nitrogen. Literature should be consulted in this regard.
2. Page 4. The accelerated durability testing (ADT) procedure should be also given in Experimental.
3. Page 7, line 207. Remove decimals from the BET surface area values.
4. Page 8, line 252. At which current density value the ORR onset potential (Eonset) was determined?
5. Page 9, lines 259-260. “As shown in Figure 4d, CoSA/N-PCMS has a lower hydrogen peroxide yield and a higher electron transfer number than Pt/C,” What could be the reason for that? It is well-known that Pt/C catalyzes a 4-electron reduction of oxygen. Literature should be consulted in this regard.
6. Since single-atom catalysts (SACs) are prone to aggregation, do they maintain their single-atom structure after electrochemical testing? The catalyst materials should be analyzed after ADT.
7. The ORR polarization curves start well below zero current density in Fig S11, the material should be stabilized first by performing several runs at fast scan rate before the RDE testing. Also, the background correction for the polarization curves should be made.
8. The authors should conduct the cyanide poisoning test to confirm the presence of CoNx in SACs.
9. How did the final catalyst have 4.28 wt% Zn in CoSA/N-PCMS as per Table S1? The zinc should have been evaporated at 1000 °C.
10. Figure S8. The O 1s XPS peak is rather large. What could be reason of that?
11. Page 2, line 38. Recent reviews are missing (https://doi.org/10.1016/j.apcatb.2022.121733; https://doi.org/10.1016/j.coelec.2023.101229). These review articles should be cited.
12. Page 2, line 46. Important literature is missing (https://doi.org/10.1016/j.electacta.2022.141676; https://doi.org/10.1016/j.jpowsour.2021.230819; https://doi.org/10.1016/j.powera.2021.100052; https://doi.org/10.1016/j.electacta.2023.142126). These papers should be cited.
Minor remarks:
Page 1, line 24. “of the ORR process.” The abbreviation of ORR should be spelled out when used for the first time in Abstract.
Page 2, line 56. ‘will-defined’ should be replaced with ‘well-defined’
Page 3, line 85. “Nafion (5%)” should be “Nafion solution (5%)” (see also page 4, line 114)
Page 4, line 118. ‘were submitted’ should be replaced with ‘were carried out’
Page 4, line 129. “(solution) in 0.1 M KOH,” should be “in 0.1 M KOH solution,”
Page 7, line 221. ‘graphite nitrogen’ should be replaced with ‘graphitic nitrogen’
Page 8, line 248. “ORR reduction peaks,” should be “ORR peaks,”
Page 8, line 254. “from LSV calculations revealed” should be “from LSVs revealed”
“Supporting Information” should be above the title of paper in Supporting Information.
Reviewer 3 Report
The work entitled “Competitive coordination-oriented monodispersed cobalt sites on N-rich porous carbon microspheres catalyst for high-performance Zn-air batteries” proposes the synthesis of Co-based N-doped carbon electrocatalysts for the oxygen reduction reaction. Based on my evaluation, I recommend the publication of the manuscript after minor revisions:
1. The authors provide information about XAS experiments, but there is a lack of references to support these values. It is necessary to add more bibliographic literature to strengthen the discussion. I suggest including the following recent references: https://doi.org/10.1002/adfm.202300405, https://doi.org/10.1016/j.nanoen.2021.106793
2. In Figure 2e, there is an error in the X-axis, and the last value should be 2500 cm-1 instead of 250 cm-1. The authors need to recheck and make the necessary corrections.
3. The authors need to provide more evidence to support their claim that the electrochemical values obtained from CoSA/N-PCMS are superior to those obtained from commercial Pt/C electrocatalysts. The LSV curves of Pt/C are significantly different from those obtained from similar Pt/C in literature, which may lead to an overinterpreatation of the electrochemical values of the prepared catalysts. It is essential to repeat and show the true Pt/C catalytic activity to validate the authors statement.
4. Regarding the EONSET, it is unclear how the authors calculate it. There are standard methods available in the literature for measuring EONSET and the authors need to refer which of these they are using. I recommend including a reference to one of these standard methods to ensure thaccuracy: https://doi.org/10.1016/j.carbon.2021.12.078
Round 2
Reviewer 2 Report
Manuscript ID: nanomaterials-2312932
“Competitive coordination-oriented monodispersed cobalt sites 2 on N-rich porous carbon microspheres catalyst for high-performance Zn-air batteries”, by Mengxia Shen, Hao Yang, Qingqing Liu, Qianyu Wang, Jun Liu, Jiale Qi, Xinyu Xu, Jiahua Zhu, Lilong Zhang, Yonghao Ni
The authors addressed all comments properly and revised the manuscript carefully. The manuscript can be accepted for publication in Nanomaterials after minor revision.
Further comments to the authors:
1. Since authors speculate that carbonating in a relatively airtight porcelain boat and the airflow cannot completely carry away the volatilized Zn, then why did the authors not carry out acid leaching using HCl or H2SO4 solution to remove the Zn from the sample before electrochemical testing? As 4.28 wt% Zn in CoSA/N-PCMS is quite high.
2. Page 4, Equation (3). B needs to be defined in Eq. (3). This is the Levich parameter.
3. Page 2, line 42. A review by Sarapuu et al. is missing (DOI: 10.1039/c7ta08690c).
4. Supporting Information, Fig. S11. The inset to Fig. S11 is too small and not clearly visible. The inset should be presented as a separate graph (Fig. S11b). The RDE plot should be Fig. S11a.
